# Unraveling the Antibacterial Mechanism of Plasma-Activated Lactic Acid against *Pseudomonas ludensis* by Untargeted Metabolomics

**DOI:** 10.3390/foods12081605

**Published:** 2023-04-10

**Authors:** Zhaobin Wang, Xiaoting Wang, Xiaowei Sheng, Luling Zhao, Jing Qian, Jianhao Zhang, Jin Wang

**Affiliations:** 1National Center of Meat Quality and Safety Control, Collaborative Innovation Center of Meat Production and Processing, Quality and Safety Control, College of Food Science and Technology, Nanjing Agricultural University, Nanjing 210095, China; 2College (School) of Food and Drug, Luoyang Normal University, Luoyang 471934, China; 3Key Laboratory of Environmental Medicine and Engineering, Ministry of Education, Department of Nutrition and Food Hygiene, School of Public Health, Southeast University, Nanjing 210009, China

**Keywords:** *Pseudomonas lundensis*, plasma-activated lactic acid, antibacterial activity, mechanism, untargeted metabolomics

## Abstract

Plasma-activated liquid is a novel non-thermal antibacterial agent against a wide spectrum of foodborne bacteria, yet fewer studies focused on its disinfection of meat spoilage bacteria. In this study, the antibacterial properties of plasma-activated lactic acid (PALA) on *Pseudomonas lundensis,* isolated and identified from spoilage beef, were investigated. A plasma jet was used to treat lactic acid (0.05–0.20%) for 60–120 s. The results presented that the 0.2% LA solution treated with plasma for 120 s caused a 5.64 log reduction. Additionally, the surface morphology, membrane integrity and permeability were altered slightly and verified by scanning electron microscopy, double staining of SYTO-9 and propidium iodide, and a K^+^ test kit. The intracellular organization of the cells, observed by transmission electron microscopy, was damaged significantly. Increased intracellular reactive oxygen species (ROS) levels exceeded the antioxidant ability of glutathione (GSH), leading to a reduction in the activity of malate dehydrogenase (MDH), succinic dehydrogenase (SDH) and intracellular ATP levels. Metabolomics analysis indicated that the energy and synthesis of essential components, such as DNA and amino acid-related metabolic pathways, were disturbed. In conclusion, this research established a theoretical basis for the use of PALA in refrigerated beef preservation by shedding light on the bacteriostatic effect of PALA against *Pseudomonas lundensis*.

## 1. Introduction

Fresh beef, which is rich in high-quality protein and fat and contains beneficial nutrients, such as niacin, vitamin B6, vitamin B12, phosphorus, potassium, iron and zinc [1,2], is susceptible to contamination and spoilage by microorganisms at any point in the food chain, from production to consumption, leading to a decrease in meat nutrition as well as unpleasant off-flavors and odors and substantial financial losses [3]. *Pseudomonas* spp. has been identified as the predominant specific spoilage organism (SSO) of chilled beef during aerobic storage [4,5]. The spoilage mechanisms of *Pseudomonas* spp. can be explained by the fact that a large amount of the colony rapidly forms at a later storage period and then produces proteinases and lipase to degrade the principal components (proteins and lipids) of fresh beef, resulting in the formation of unpleasant odors and mucous substances on the surface of meat [6]. *Pseudomonas* spp., like other Gram-negative bacteria, has a lipopolysaccharide layer on its outer membrane that acts as a permeability barrier, making it less vulnerable to antimicrobials than Gram-positive bacteria [7]. It is crucial to develop antibacterial agents that effectively develop antibacterial agents that efficiently block the development of spoilage or pathogenic bacteria without changing the taste or chemistry of the food. In meat factories, conventional antibacterial sanitizers use lactic acid spray-washing on beef carcasses at low temperatures [8]. Samelis et al. [9] and Deumier [10] both demonstrated that although lactic acid (2–5%) significantly reduced surface contamination and extended the shelf-life of meat, a high concentration (>1%) could negatively affect the quality of meat.

A recent candidate for disinfection includes plasma-activated water (PAW), which is generated by treating water with non-thermal plasma that is activated by various sources, including pulsed coronas, dielectric barrier discharges [11,12] and atmospheric pressure plasma jets [13]. Compared with other emerging non-thermal technologies, PAW, made using a plasma jet, is characterized by easy operation, no-residue and high efficiency for microbial inhibition in heat-sensitive food [14,15]. Prior research has linked the production of reactive oxygen species (ROS) and reactive nitrogen species (RNS) [16], such as OH radicals, superoxides and nitric oxide, to the potent antibacterial action of PAW [17,18]. When these reactive species dissolved in liquids, the long-lasting antibacterial effects of PAW were caused by the formation of hydrogen peroxide (H_2_O_2_), nitrite (NO_2_), nitrate (NO_3_) and peroxynitrites (ONOO^−^) anions [19,20].

Lactic acid was dissolved in water prior to the plasma treatment in order to increase the microbial inhibitory efficiency of PAW. By decreasing the pH of PAW, lactic acid promotes the production of NO_2_^−^ and H_2_O_2_ [21], and the reaction between NO_2_^−^ and H_2_O_2_ produces peroxynitrous acid (ONOOH), which is toxic to cells [22]. Qian et al. [23] found that plasma-activated lactic acid (PALA) at 0.20% possessed virtually the same antibacterial power of *S. enteritids* as 2% lactic acid therapy without any apparent negative effects on the beef properties, which indicated that PALA has the potential to be a safe and effective decontaminant agent.

The objectives of this experiment are (1) to assess the antibacterial efficiency of plasma-activated lactic acid on *Pseudomonas* spp., isolated and identified from chilled spoilage beef, and (2) to examine the morphophysiological, oxidative stress response (intracellular ATP level, GSH content) and energy metabolism change in *Pseudomonas* spp. when exposed to PALA. This research provided theoretical support for a promising alternative sanitizer in the beef sector by elucidating the antibacterial mechanisms of beef deterioration bacteria under PALA treatment.

## 2. Materials and Methods

### 2.1. Microbial Analysis

#### 2.1.1. Preparation of Plasma-Activated Lactic Acid Solution

A non-thermal atmospheric-pressure plasma jet was used to prepare the PALA solution, as described by Qian et al. [24]. Concentrations of 0.05%, 0.10%, 0.15% and 0.20% of lactic acid (LA) in PALA were prepared by adding 150, 300, 450 and 600 μL of LA to 300 mL of deionized water (DIW), respectively. The aforementioned solutions were subjected to the plasma jet treatment for 0, 60, 80, 100 and 120 s, with the terminal of the plasma jet positioned approximately 2 cm below the surface of the solution. The DIW and PAW were chosen as control groups. PALA (0.05%), PALA (0.10%), PALA (0.15%) and PALA (0.20%) represented the PALA solutions with the corresponding LA concentrations, respectively.

#### 2.1.2. Bacterial Culture and Antibacterial Activity of PALA

Bacteria were collected from rotted beef carcasses in a local beef processing facility. *Pseudomonas. lundensis* (PZ3) was isolated and identified by combining 16S DNA sequences and typical biochemical reactions and storing them at −80 °C for the next use. Prior to each experiment, the cultures were activated on Pseudomonas CFC selective Agar (CFC, Hope- Biotechnology, CO. Ltd., Qingdao, China) at 28 °C for 24 h. A single colony was selected and cultured in 6 mL of Luria-Bertani (LB, Hope-Biotechnology, CO., Ltd. Qingdao, China) broth by shaking at 180 rpm at 28 °C for 10 h. After three washes in sterile 0.85% NaCl solution, the bacterial suspensions were centrifuged at 5000× *g* at 4 °C for 7 min to remove debris. The concentration of cells was adjusted to nearly 10^8^ CFU/mL.

Before incubating at room temperature for 10 min, a 1 mL aliquot of the cell suspension was well mixed and vortexed with 1 mL of PALA with different LA concentrations (0.05–0.20%). Thereafter, serial 10-fold diluted solutions (1 mL) were spotted onto CFC agar, respectively. Incubation of the samples at 28 °C for 24 h yielded the final cell counts, which were then reported as log colony-forming units per milliliter (log CFU/mL).

### 2.2. Morphological Observation of Pseudomonas lundensis

#### 2.2.1. Scanning Electron Microscopy

After treating *P. lundensis* with PALA (0.20%), we observed its morphology change using scanning electron microscopy (SEM) in the same way described by Wang et al. [25]. Briefly, after the exposure with PALA (0.20%) and DIW separately for 10 min at room temperature, *P. lundensis* strains were fixed by 2.5% glutaraldehyde overnight, then washed with 0.1M phosphate-buffered saline (PBS, pH 7.2), dehydrated with a series of ethanol gradients (50%, 70%, 80%, 95%, 100%) and resuspended in tert-butanol. All samples were sputter-coated with 10 nm thick gold particles before being photographed with a Hitachi Se3000N scanning electron microscope (Hitachi, Tokyo, Japan).

#### 2.2.2. Transmission Electron Microscopy (TEM)

The samples were pre-treated in the same way as for SEM, and then transmission electron microscopy (TEM) was used to examine the structural effects of PALA on *P. lundensis.* The cultures were precipitated by centrifugation, washed twice with 0.1 M PBS (pH 7.2), and then centrifuged. The obtained precipitants (the control and treatment groups) were fixed in 2.5% glutaraldehyde overnight, paraffin-embedded, sliced, and doubly stained with 3% uranium acetate and lead nitrate before being viewed using a Hitachi H7700 TEM. As a control, we tested the bacteria not exposed to PALA.

#### 2.2.3. Measurement of Membrane Integrity

Following the manufacturer’s instructions, we used a live/dead detection kit (Beijing Solarbio Science & Technology Co., Ltd., Beijing, China) to test the cell membrane integrity viability of *P. lundensis* cells after exposure to PALA. The components of the kit are SYTO-9 and Propidium iodide (PI). Bacteria with a concentration of around 10^8^ CFU/mL were centrifuged at 5000× *g* for 10 min while exposed to PALA for 10 min. The gathered precipitants were washed and resuspended in binding buffer and stained with 200 μL of SYTO-9 and PI at room temperature in the darkness, respectively. A confocal laser scanning microscope was used to take pictures of the samples (FluoView FV1200 OLYMPUS, Tokyo, Japan).

#### 2.2.4. Measurement of Leakage of K^+^

The cell permeability of *P. lundensis* was identified by measuring extracellular K^+^. After centrifugation at 10,000× *g* for 10 min, the supernatant from both the control and PALA-treated cell suspensions were collected and examined. The concentration of K^+^ was examined by following the guidelines of the K^+^ detection kit (Beijing Solarbio Science & Technology Co., Ltd., Beijing, China). The result was shown as the leakage rate = K^+^ (concentration of PALA)/K^+^ (concentration of Control) (*n* = 3).

### 2.3. Measurement of Intracellular ATP Concentration

Intracellular ATP levels were determined using a commercially available ATP test kit (Beyotime Institute of Biotechnology, Beijing, China). The luminescence signal in this test was directly proportional to the concentration of ATP. *P. lundensis* was cultured in the lab and collected at a concentration of 10^8^ CFU/mL, as was previously described. After treatment, 1 mL of bacteria suspension was centrifuged at 10,000× *g* for 6 min, and the pellets were mixed with 200 μL of lysis buffer from the kit and re-centrifuged. The supernatant was transferred to a fresh tube, and the protein concentration was adjusted using a protein assay kit (Beyotime Institute of Biotechnology, Beijing, China). A 100 μL sample and detection buffer were mixed and then assayed in a luminometer (SpectraMax M2e; Molecular Devices, San Jose, CA, USA) to measure the luminescence.

### 2.4. Measurement of Intracellular ROS and Glutathione

2′,7′-dichlorodihydrofluoresecinc (DCFH) (Solarbio, Beijing, China) is a cellular test probe commonly used for the detection of intracellular ROS, which may induce oxidative damage in bacteria and serve as a signal of oxidative stress [26]. Cells before and after the PALA treatment were collected by centrifugation at 5000× *g* for 7 min, washed 3 times with 0.1 M PBS (pH 7.2) and discarded in the supernatant. The pellets were incubated with 1 mL of 10 μ mol/L DCFH-diacetate (DA) (Sigma-Aldrich Ltd., Dublin, Ireland) for 15 min at 37 °C, then washed with 0.1 M PBS (pH 7.2) again. Finally, the cells were transferred into a 96-well fluorescence microplate and measured by a microplate reader (SpectraMax M2e, Molecular Devices, Radnor, PA, USA) at the excitation and emission wavelengths of 485 and 525 nm, respectively.

The amount of glutathione (GSH) contained within the cells was determined using a glutathione (GSH) assay kit purchased from the Beyotime Institute of Biotechnology in China. Briefly, after the bacteria concentration of 10^8^ CFU/mL was obtained, as previously described, the cell pellets were washed with 0.1M PBS (pH 7.2) two times and then centrifuged, discarding the supernatant. After adding and thoroughly vortexing three volumes of protein scavenging reagents M, the mixture was frozen twice in liquid nitrogen and thawed twice in a 37 °C water bath. The supernatants were collected to measure total glutathione (T-GSH) and oxidized disulfide (GSSH) via the auto-microplate reader at 412 nm. The concentrations of GSH were obtained by the reduction of GSSH levels from the T-GSH (GSH = T-GSH-2 × GSSH).

### 2.5. Detection of the Intracellular Activity of MDH and SDH

Because of their importance in the tricarboxylic acid (TCA) cycle, the activities of the Krebs cycle enzymes, malate dehydrogenase (MDH) and succinic dehydrogenase (SDH) were measured as a proxy for *P. lundensis*’s oxidative metabolic capability. In this research, the activities of these two enzymes were measured according to the guidelines of the MDH and SDH viability assay kits (Beijing Solarbio Science & Technology Co., Ltd, Beijing, China). The protein concentration level was determined using the BCA method. The activity unit (U/mg prot) of MDH and SDH were, respectively, expressed as 1 nmol NADH and 1 nmol 2,6-dichloroindophenol, consumed by 1 mg of tissue protein in the reaction system within 1 min.

### 2.6. Metabolomics Analysis

#### 2.6.1. Sample Preparation and Metabolites Extraction

An untargeted metabolomics strategy was used to examine how PALA might alter the metabolites and metabolic pathways of *P. lundensis*. Bacterial sludge was prepared by centrifuging the cultures of test bacteria at 7000× *g* to remove the supernatant, reaching a final concentration of 10^8^ CFU/mL in the centrifuge tube. The sludge was double-washed with a saline solution containing 0.85% NaCl. PALA (0.20%) was added to a bacterial solution of the same volume, which was then properly mixed, subjected to a 10-minute reaction time, and centrifuged at 4000× *g* for 5 min. Deionized water was used as a placebo in the control group. There were four cycles for both the treatment and control groups. Both treatment and control groups were implemented 4 times.

Furthermore, 100 mg bacteria sludges were grounded with liquid nitrogen, and the homogenate was resuspended with 400 μL of prechilled 80% methanol and 0.1% formic acid. The samples were centrifuged at 15,000× *g* at 4 °C for 5 min after being incubated on ice for 5 min. The precipitants were then resolubilized with 100 μL of 53% methanol, sonicated for 6 min at 4 °C, and centrifuged at 15,000× *g* at 4 °C for 5 min. The last step was to inject the supernatant into the LC-MS/MS system.

#### 2.6.2. UHPLC-MS/MS Analysis

The Vanquish UHPLC system (Thermo Fisher, Bremen, Germany) and Orbitrap Q Exactive^TM^ HF mass spectrometer were used for the UHPLC-MS/MS analysis (Thermo Fisher, Bremen, Germany). The samples were injected at a flow rate of 0.2 mL/min onto a 17-minute linear gradient on a Hypesil Gold column (100 × 2.1 mm, 1.9 μm). The eluents for the positive polarity and negative mode were set as eluent A (0.1% formic acid) and eluent B (Methanol), and eluent A (5 mmol/L ammonium acetate, pH 9.0) and eluent B (Methanol) separately. The solvent gradient was set as follows: 2% B, 1.5 min; 2–100% B, 12.0 min; 100% B, 14.0 min; 100–2% B, 14.1 min; 2% B, 17 min. The Q Exactive^TM^ HF mass spectrometer was operated in the positive/negative polarity mode with a spray voltage of 3.2 kV, a capillary temperature of 320 °C, a sheath gas flow rate of 40 arb and an aux gas flow rate of 10 arb.

#### 2.6.3. Data Processing and Metabolite Identification

Compound Discover 3.1 (CD3.1, Thermo Fisher) was used to carry out the peak alignment, peak selection and quantification for each metabolite from the raw data files obtained by UHPLC-MS/MS. The following served as the primary determinants: retention time tolerance, 0.2 min; actual mass tolerance, 5 ppm; signal intensity tolerance, 30%; signal/noise ratio, 3; and minimum intensity, 100,000. Afterward, the spectral intensity was normalized to the peak intensity. Additive ions, molecular ion peaks and fragment ions were used to infer the molecular formula from the normalized data. Accurate and relative quantitative results were obtained by matching the peaks to the mzCloud (https://www.mzcloud.org/, (accessed on 10 February 2023)). Afterward, the spectral intensity was normalized to the peak intensity. Additive ions, molecular ion peaks and fragment ions were used to infer the molecular formula from the normalized data. Accurate and relative quantitative results were obtained by matching the peaks to the mzCloud. These metabolites were annotated using the KEGG database (https://www.genome.jp/kegg/pathway.html, accessed on 10 February 2023), HMDB database (https://hmdb.ca/metabolites, accessed on 10 February 2023) and LIPID Maps database (http://www.lipidmaps.org/, accessed on 10 February 2023). Principal component analysis (PCA), as well as partial least squares discriminant analysis (PLS-DA), were performed using metaX (a flexible and comprehensive software for processing metabolomics). To determine the statistical significance (*p*-value), we used a one-way analysis of variance (*t*-test). The metabolites with variable importance in the projection (VIP) of > 1, a *p*-value < 0.05 and a fold change (FC) of ≥2 or ≤0.5 were considered to be differential metabolites. The Pheatmap tool in R was used to generate the clustered heat maps; the data was normalized using z-scores of the intensity areas of differential metabolites. Cor. in R language (method = Pearson) was used to investigate the statistical correlation between distinct metabolites. The R package cor.mtest was used to determine the statistical significance between different metabolites. Correlation plots were generated using the corrplot package in the R programming language, and a *p*-value of less than 0.05 was considered statistically significant. The KEGG database was used to investigate the roles of various metabolites and metabolic pathways. When the ratio x/n > y/N was reached, we defined the metabolic route as enriched. The metabolic pathways were considered statistically significant enrichment when the *p*-value of the metabolic pathway was <0.05.

### 2.7. Statistical Analysis

The data are shown as a mean ± standard deviation (SD). SPSS (Version 22, IBM, Armonk, NY, USA) was utilized for the statistical analysis, and ANOVA was employed to determine the statistical significance between groups. All experiments were performed using 3 replicates, excluding the metabolomics analysis, which was conducted in 4 replicates.

## 3. Results and Discussion

### 3.1. Microbial Analysis

In this study, we examined how different treatment times and lactic acid concentrations affected PALA’s antibacterial efficacy against *P. lundensis*. As shown in Figure 1, the inactivation efficacy of the *P. lundensis* cells notably increased in LA concentration-dependent and treatment time-length-dependent manners. After exposure to PAW and PALA, the number of live bacteria dropped dramatically, and this trend held true when the amount of LA added remained the same, indicating that a plasma treatment for 120 s demonstrated the strongest antibacterial effect. From an initial concentration of 8.12 log CFU/mL, the number of live bacteria reduced to 5.88, 4.21, 3.61, 3.13 and 2.48 log CFU/mL for 0, 0.05, 0.10, 0.15 and 0.20% of the PALA solution, respectively, after 120 s of treatment. However, no significant difference in antibacterial ability was observed between 100 s and 120 s of the PALA (0.20%) treatments, and the number of viable cells was reduced by nearly 5.64 log CFU/mL. Several researchers have investigated the antibacterial activity of other plasma-activated solutions. Wu et al. [27] found that treating *S. aureus* suspensions with plasma-activated hydrogen peroxide for 10 min resulted in only a 4.6 log CFU/mL reduction, even when the ratio of the plasma-activated solution to *S. aureus* was as high as 99:1. Moreover, it took 15–30 min for 10 mL of plasma-activated water to cause a 3.1–4.4 log CFU/mL reduction of *Leuconostoc. Mesenteroides* [28]. PALA is a possible sanitizer for the meat industry since its antibacterial effectiveness and antibacterial ability are greater than those of PAW and other plasma-activated solutions.

### 3.2. Surface Morphology and Plasma Membrane Integrity

A SEM inspection and an SYTO-9/PI double staining experiment were used to examine the surface ultrastructure and cell membrane integrity of *P. lundensis* to evaluate the method of action of PALA on the morphology. Figure 2A,B indicate that the DIW-treated cells exhibit distinct rod-shaped and smooth cell membranes of *Pseudomonas.* spp. In contrast, for bacteria after the PALA (0.20%) treatment (Figure 2C,D), partial shrinkage and some holes appeared on the membrane surface, indicating that PALA only exerted a slight morphological impact on the cells. It matched the LSCM pictures of DIW, PAW, 0.1% PALA, and 0.2% PALA very well (Figure 3). Dye PI was used to identify bacteria with damaged membranes, while SYTO-9 was used to identify viable bacteria in order to study membrane integrity. The higher concentration of PALA caused a constant increase in the PI-stained cells. However, the percentage of SYTO-9 was nearly 10 times that of the PI-stained cells, even at PALA (0.2%), which indicated that only a small percentage of the cell’s membrane integrity was damaged due to the PALA treatment.

A number of researchers have studied the effect of plasma on the shape of bacterial membranes. Han et al. [29] discovered that *E. coli* experienced breaking effects and substantial cell shrinkage under 1 min high-voltage atmospheric cold plasma (ACP) treatment. Zhao et al. [30] treated *S. aureus* with a plasma-activated hydrogen solution (PAH) for 30 min and found that PAH-treated cells revealed obvious cytoplasm leakage and severe deformation when compared to a normal cell shape. The inconsistent degree of membrane damage, compared with our results, might be explained by the variation in treatment time length, the source of plasma and the membrane structure of the target bacteria chosen.

### 3.3. Effect of PALA on Intracellular Organization

Transmission electron microscopy (TEM) was used to directly evaluate the intracellular organization change that was induced in the *P. lundensis* cells upon exposure to PALA. The TEM images clearly showed that the intracellular organization of the cells incubated with PALA (0.2%) was obviously damaged (Figure 4). As Figure 4A,B reveals, the untreated cells displayed a uniform cytoplasm, good integrity and evenly distributed condensed electron-transparent regions, representing bacterial nucleoids. Following treatment with PALA (0.2%) for 10 min (Figure 4C,D), the phenomena of the partial separation of the cell wall and membrane, an obviously lower density of cytoplasm and nucleic acid, and the destroyed integrity of the cell wall structure were observed. However, relatively fewer leaked materials could be found around the cells’ surroundings. These observations demonstrated that PALA mainly induced severe damage to the intracellular organization of cells. Combined with the SEM image, a hypothesis that the intracellular organization of *P.lundensis* is more susceptible to PALA than the cell membrane and that intracellular organization alterations were more severe than the morphological change could be made [31].

### 3.4. Intracellular ROS Level and GSH Concentration

We also measured the intracellular reactive oxygen species (ROS)—which is widely regarded as the major inactivation agent in PAW [32]—to verify if oxidative stress had affected the bacterium cells from the inside. The intracellular ROS levels of all plasma-treated samples were significantly higher than the control (Figure 5a). The increase in the concentration of intracellular ROS was dependent on the LA concentration of PALA (0–0.15%), which indicated that PALA caused elevated cellular oxidative stress inside the cells. When the LA concentration in PALA was up to 0.15%, the intracellular ROS levels achieved the highest and then slightly decreased when the LA concentration in PALA was 0.2%. This was likely because the oxidative stress induced by PALA (0.20%) went beyond the ROS-scavenging system within cells, thereby altering membrane permeability and resulting in the leakage of intracellular ROS [33].

GSH plays multiple roles in protecting against environmental stresses, such as osmotic shock and acidity, as well as against toxins and oxidative stress in Gram-negative bacteria [34]. As shown in Figure 5b, the GSH concentration showed a rising trend and then decreased with the increased LA concentration in PALA. At first, the GSH levels in PAW and PALA (0.05%) were 1.32 and 1.27 times greater than in CK (DIW), respectively, indicating that an abundance of ROS within the bacteria prompted a rise in the GSH levels to counteract the oxidative damage they were experiencing. The intracellular GSH content dropped with an increasing LA concentration (*p* < 0.05), with a particularly dramatic decline following the treatment with PALA (0.1%), which was 0.71 times the CK content. Then, for the bacteria incubated with PALA (0.20%), the GSH concentration was lower by 0.06 times the CK content. These results suggest that the oxidative stress levels exceeded the GSH’s defense capacity for LA concentrations in PALA over 0.1%, consequently accelerating the death of cells [33,35].

### 3.5. Leakage Ratio of K^+^, Intracellular Contents of ATP and MDH, and SDH Activity

The effect of PALA on membrane permeability was evaluated by the leakage of K^+^. As Figure 6a shows, the addition of PAW and PALA (0.05%) induced a slight leakage of K^+^, and then gradually increased to 1.31 times the CK content (DIW) when the LA concentration of PALA was 0.2%, which is in good agreement with the SEM images that partially demonstrated shrinkage but not breakage of the membrane following PALA being added, indicating that PALA gently modified the permeability of the cell membrane, thus causing leakage of small ions, such as K^+^ [36].

To investigate the effect of PALA on the energy metabolism of *P. lundensis,* the intracellular ATP concentration and activities of MDH and SDH were measured. The effect of PALA on the intracellular ATP concentration of the cell suspensions is shown in Figure 6b. The intracellular ATP concentration gradually decreased (*p* < 0.05) with increased LA concentrations, resulting in a decrease from 2.12 nmoL/mg at control to 0.05 nmoL/mg under 0.2% LA of PALA. The SDH activity showed a continuous decrease trend as increasing LA concentrations (0–0.20%) of PALA occurred (Figure 6c), which were 0.78, 0.52, 0.27, 0.16, and 0.06 times that of CK. The activity of MDH first increased when treated with PAW, then gradually decreased in an LA-concentration-dependent manner (Figure 6d). After the PALA (0.20%) treatment, it reached the lowest level, which was 0.06 times that of CK. PALA interfered with the energy metabolism in *P. lundensis* by reducing MDH and SDH activity and ATP synthesis. Similar results were reported by Liao et al. [37], who reported that plasma-activated solutions decreased the intracellular ATP concentration in a concentration-dependent manner.

MDH and SDH are key dehydrogenases for ATP production through a tricarboxylic acid cycle and oxidative phosphorylation pathway in prokaryotic cells. Oxalacetic acid (OAA) to malate is a reversible reaction that is linked with the oxidation/reduction of dinucleotide coenzymes, and MDH can catalyze this reaction [38]. SDH catalyzes the dehydrogenation of succinic acid to fumarate, which will provide electrons for a variety of respiratory chains of prokaryotic cells [39]—the creation of proteins and other metabolic processes that require energy drawn from ATP. PALA therapy may have depleted cytoplasmic ATP because of decreased ATP synthesis or increased ATP hydrolysis by the proton-pump ATPase as a result of an accumulation of reactive oxygen species within the cell [40]. Cai et al. [41] proposed that AEW treatment decreasing the intracellular ATP concentration might be partially explained by the enhanced cell membrane permeability and membrane impairment under AEW, leading to leakage of ATP and an altered balance of cations, such as H^+^ and Na^+^. Our previous experiments showed that membrane completeness and permeability were slightly altered, and the leakage rate of small ions, such as the K^+^ of cells, was much lower when compared to a reduction in the ATP concentration when cells were exposed to the PALA (0.20%) treatment. Thus, alterations in ATP-related activity, such as the reduction activities of MDH and SDH but not membrane alterations, play a major role in the depletion of intracellular ATP of *P. lundensis* following PALA treatment.

### 3.6. Metabolomic Analysis

Non-targeted metabolomics was applied to identify the key metabolic pathways that caused bacterial death and reveal the mechanism of bacterial inhibition. The *P. lundensis* metabolic changes in response to the PALA treatment were profiled using multivariate statistical models, PCA and PLS-DA. As depicted in Figure 7A,B, the PC1 variable in the principal components in the negative and positive modes explained the variations of 85.4% and 83.2% of the original data, respectively. PCA analysis showed that PALA caused significant changes in the intracellular metabolites of bacteria. Partial least squares discriminant analysis (PLS-DA) was applied to obtain the maximal separation between profiles and could verify the metabolites mainly in charge of discrimination in the pairwise comparison [42]. The PLS model in the present study (Figure 7C) showed high R^2^ and Q^2^ values (0.996 and 0.999, respectively) and thus displayed a good fit and satisfactory predictive ability, which was consistent with the PCA model and further proved that PALA significantly interfered with the intracellular metabolite pathways.

The changes in the intracellular metabolites with DIW and PALA treatment were visualized and profiled with a constructed heat map (Figure 8). According to the selected criteria of differential metabolites with a VIP of >1, *p*-value of <0.05 and a fold change (FC) of ≥ 2, a total of 131 differential metabolites were identified, including 6 upregulated metabolites and 125 downregulated metabolites (Appendix A).

The KEGG database was applied to divide the identified differential metabolites into different categories according to the metabolic pathway and function. The key pathways that had a high correlation due to the metabolite differences caused under PALA (0.20%) were screened by the KEGG enrichment pathway analysis. The disordered metabolites were mainly enriched in purine and pyrimidine metabolism, the ABC transporters pathway, the biosynthesis of amino acids, and pyrimidine metabolism (Figure 9). Furthermore, the citric cycle pathway, oxidative phosphorylation, pyrimidine metabolism, glutathione (GSH) metabolism, and glycerophospholipid metabolism are closely related to energy metabolism and membrane damage caused by excessive ROS stress and were also investigated.

The hypothesis that PALA slightly affected the membrane integrity and permeability, and therefore cell death under PALA, could not be ascribed mainly to the morphological damage from the above results and was also investigated using metabolomics. The membrane functional and structural changes were highly related to the metabolic profiles of lipids, including phospholipids, glycolipids and fatty acids [43]. Of note, the cytoplasmic membrane of *P. lundensis* comprises saturated fatty acids (palmitic acid, stearic acid, etc.) and unsaturated fatty acids (7-hexadecenoic acid, cis-10-heptadecenoic acid, etc.), and a certain ratio of saturated fatty acids and unsaturated fatty acids are responsible for the fluidity and deformability of cells [44]. No significant changes in these fatty acids were observed (*p* > 0.05) (Appendix A), which indicated that excessive ROS resulting from PALA stimulation unremarkably caused the lipid oxidation of unsaturated fatty acids. Furthermore, glycerophospholipids, as membrane lipids, not only functioned as a component of the lipoproteins but also contributed to the regulation of the transport processes, protein function and signal transduction [45]. As an intermediate product of glycerol phospholipid metabolism, the metabolite level of D-glycerol-1-phosphate slightly decreased (VIP < 1) in the PALA group compared to the control group, which could partially explain the altered cell membrane permeability.

Purine metabolism was the most relevant metabolic pathway affected by PALA exposure (Figure 10A). The metabolic analysis showed that the contents of adenosine, xanthosine, inosine, 2′-deoxyinosine, guanosine, 2′-deoxyadenosine 5′-monophosphate, xanthine, adenosine 5′-monophosphate, guanosine monophosphate and hypoxanine were significantly downregulated (*p* < 0.05) after PALA exposure. Moreover, the pyrimidine-related metabolites uridine, 2′-deoxycytidine, UMP, cytidine, 5′-monophosphate-N-acetylneuraminic acid, and cytosine were also significantly lowered (*p* < 0.05). Among these altered metabolites, uridine is one of the four bases that make up RNA. Guanosine, Hypoxanthine, xanthine and uric acid are involved in purine metabolism, in which guanine is the base for DNA and RNA metabolism [46]. DNA produces RNA by transcription, while RNA produces proteins by translation. In contrast, RNA can also generate DNA by reverse transcription. The results showed that after the PALA treatment, the contents of these five metabolites significantly decreased, which proved that PALA disturbed the synthesis of nucleic acid (DNA and RNA) and further affected the transcription, reverse transcription and translation of cells.

Moreover, the biosynthesis of amino acids was conducted. In the present study, the contents of several essential amino acids and intermediate products, including L-glutamic acid, L-tyrosine, L-methionine, L-arginine, S-adenosyl-L-methionine, S-adenosylhomocysteine and N-acetylornithine were notably downregulated (*p* < 0.05), which indicated that PALA disturbed the biosynthesis of several essential amino acids and denaturing proteins. As an intermediate product of alanine, aspartate and glutamate metabolism (Figure 10D), glutamate is generally considered a protective metabolite in cells, plays a crucial role in various stress responses [47] and shows a much lower value. Therefore, the reduced glutamic acid level in the treatment group was considered to be a loss of ability to protect cells from excessive ROS stress. Furthermore, some ABC transporter-related metabolites that play a major role in the uptake of phosphate and amino acid across the lipid bilayers [48], including glutamate, L-hydroxyproline and aspartate, were decreased (*p* < 0.05) in the treatment group, therefore disturbing the biosynthesis of arginine and the metabolism of alanine, aspartate and glutamate.

Both the GSH and GSSH contents were significantly downregulated (*p* < 0.05) (Figure 10B), which was in high accordance with the above results measured with the glutathione (GSH) assay kit, which further proved the speculation that ROS generation prevails over the capacity of the antioxidant defense system. The metabolic profile of the TCA cycle (Figure 10C) and oxidative phosphorylation (Figure 10E) have also been observed to differ significantly under PALA stress. The obtained results demonstrate that PALA dysregulated the metabolites related to the TCA cycle, such as citrate, succinate and cis-aconitate. Among these intermediates, citric acid was the product of the first irreversible reaction, formed by the catalyzed condensation of acetyl-coenzyme A (CoA) and oxaloacetic acid with citrate synthase [49]. Succinic acid was dehydrogenated and catalyzed with SDH to fumaric acid. MDH catalyzes a dehydrogenation reaction from malate to generate oxaloacetate, accompanied by the reduction of NAD to generate NADH [50]. Combined with the decreased activity of SDH and MDH, a conclusion that PALA effectively affects the *P. lundensis* TCA cycle can be made. However, in prokaryotes, most of the energy released by the TCA cycle is stored in reducing coenzymes; the energy in reducing coenzymes needs to be synthesized into ATP by oxidative phosphorylation to ensure the accomplishment of almost all the other vital processes. During oxidative phosphorylation, electrons derived from NADH and succinate are combined with O_2_, and the energy released from this oxidation/reduction reaction is used to drive the synthesis of ATP from ADP [51]. To further investigate how energy metabolism is affected by PALA, oxidative phosphorylation-related metabolites were also researched. As shown in Figure 10E, the results demonstrated a notable decrease in NADH and NAD^+^, which indicated that a lower level of electrons derived from the dehydrogenation reaction (NADH oxidized to NAD^+^) bypassed the electron transport chain to create a proton gradient across the cell membrane. ATP synthesis, utilized by complex V (ATP synthase) through this electrochemical gradient, was severely disrupted. Therefore, perturbation of oxidative phosphorylation and TCA metabolism could explain the previously significant lower value of intracellular ATP after PALA exposure.

## 4. Conclusions

To conclude, the sensitivity of *P. lundensis* cells to plasma-activated lactic acid and the underlying antibacterial mechanism were extensively analyzed. For optimal antibacterial efficacy against *P. lundensis*, lactic acid was added to distilled water following plasma treatment. PALA caused significant oxidative stress because the main intracellular antioxidant, GSH, could not handle the flood of ROS it created. The high levels of reactive oxygen species slightly affected the surface shape and plasma membrane integrity relatively. The intracellular ATP level was significantly lowered under PALA treatment and might be attributed to the decreased ATP-synthase enzyme activity (MDH and SDH). The results of untargeted LC-MS-based metabolomics proved the above results and pointed out that oxidative damage led to the disturbance of several synthesis and energy-related metabolisms, including the TCA cycle and oxidative phosphorylation, amino acid metabolism, pyrimidine and purine metabolism.

## Figures and Tables

**Figure 1 foods-12-01605-f001:**
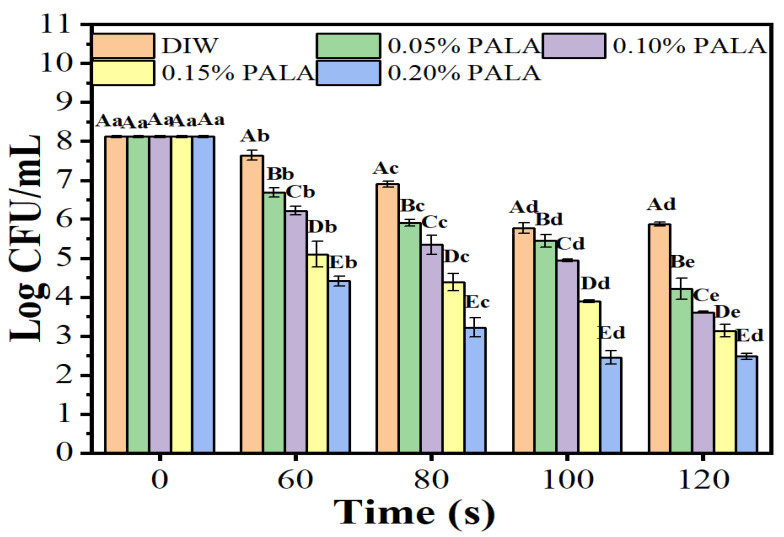
The number of viable cells of *P. lundensis* incubated for 10 min with deionized water (DIW) and 0.05–0.20% lactic acid (LA) treated with plasma for 0, 60, 80, 100 and 120 s. Error bars represent standard deviation (*n* = 3). Different uppercase letters (A–E) indicate a significant difference (*p* < 0.05) at the same treatment time over different LA concentrations of PALA. Different lowercase letters (a–e) indicate a significant difference (*p* < 0.05) at the same LA concentrations among different plasma jet treatment times with DIW.

**Figure 2 foods-12-01605-f002:**
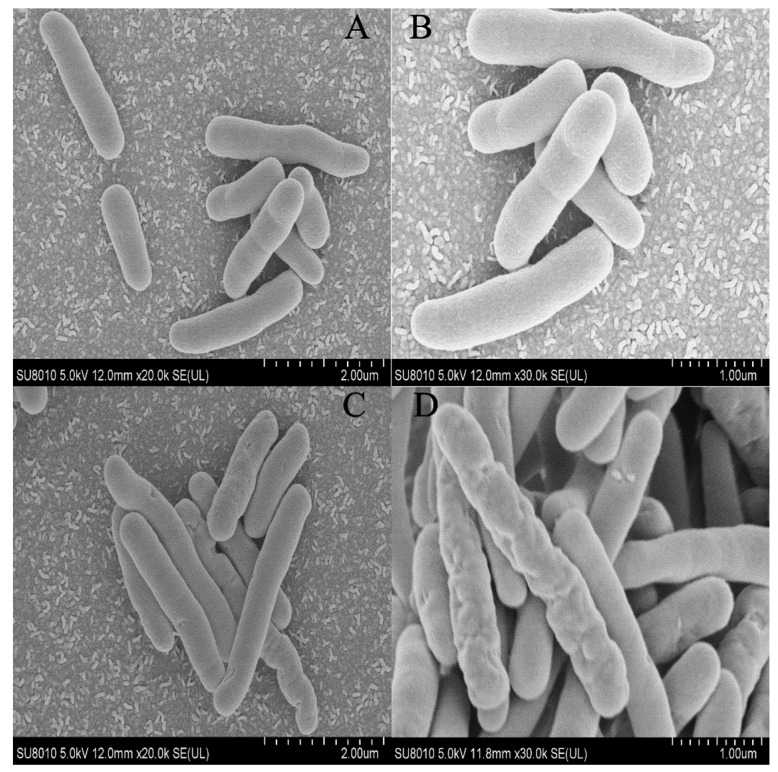
Scanning electron micrographs of *P. lundensis* cells after (**A**,**B**) DIW, (**C**,**D**) PALA (0.20%) treatment. The scan bars of 2(**A**) and 2(**C**) are 2 μm, and 2(**B**) and 2(**D**) are 1 μm.

**Figure 3 foods-12-01605-f003:**
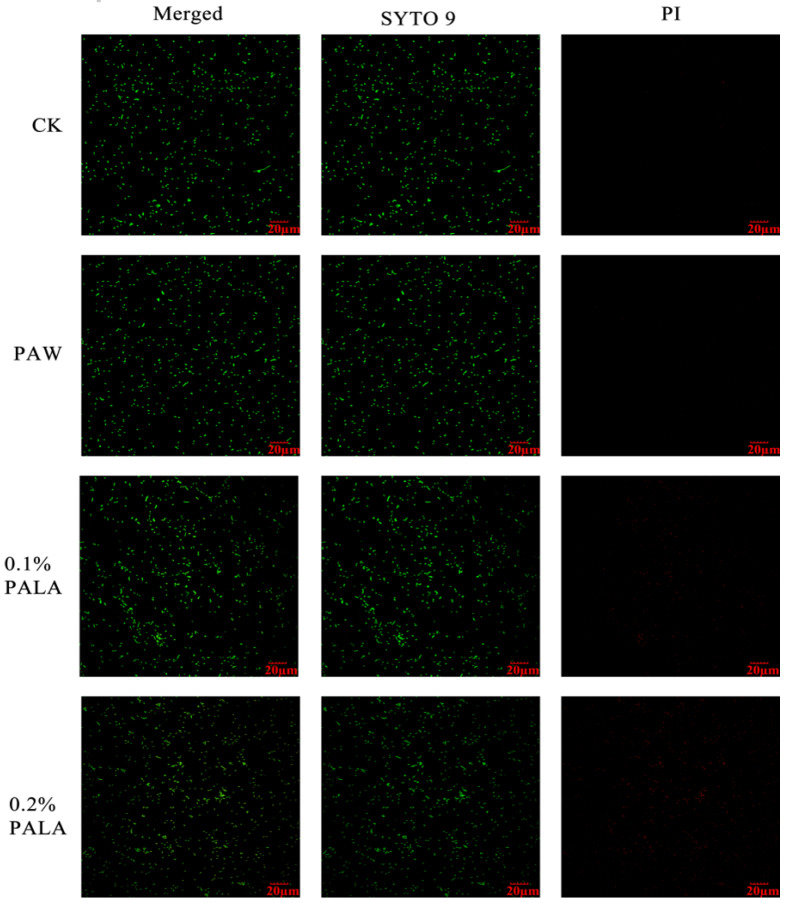
LSCM images of bacteria after DIW, PAW, PALA (0.10%) and PALA (0.20%) treatment. The scan bar is 20 μm. Each row shows Merged, SYTO-9 (green fluorescing dye) and PI (red fluorescing dye) stained samplec DIW: deionized water; LA: lactic acid; PAW: plasma-activated water; PALA: plasma-activated lactic acid.

**Figure 4 foods-12-01605-f004:**
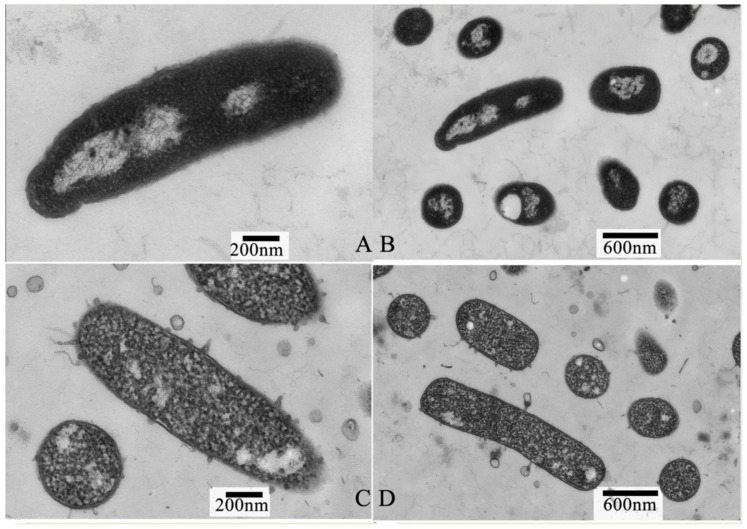
Transmission electron micrographs of *P. lundensis* cells treated with DIW (**A**,**B**) and cells treated with PALA (0.20%) (**C**,**D**). The scale bars of 4 (**A**) and 4 (**C**) are 200 μm, the scan bars of 2 (**A**) and 2 (**C**) are 200 nm and 2 (**B**) and 2 (**D**) are 600 nm.

**Figure 5 foods-12-01605-f005:**
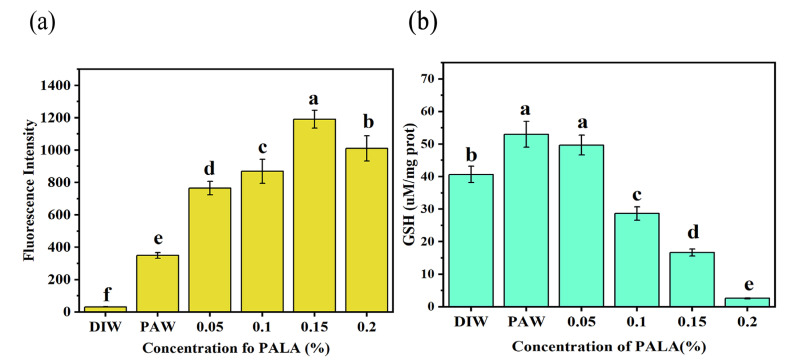
Effect of treatment with DIW, PAW, PALA (0.05%), PALA (0.1%), PALA (0.15%) and PALA (0.20%) on the intracellular ROS levels (**a**) and GSH concentration (**b**) of *Pseudomonas Lundensis*. Bars represent standard deviations (*n* = 3). a–f: values with different lowercase letters significantly differ (*p* < 0.05).

**Figure 6 foods-12-01605-f006:**
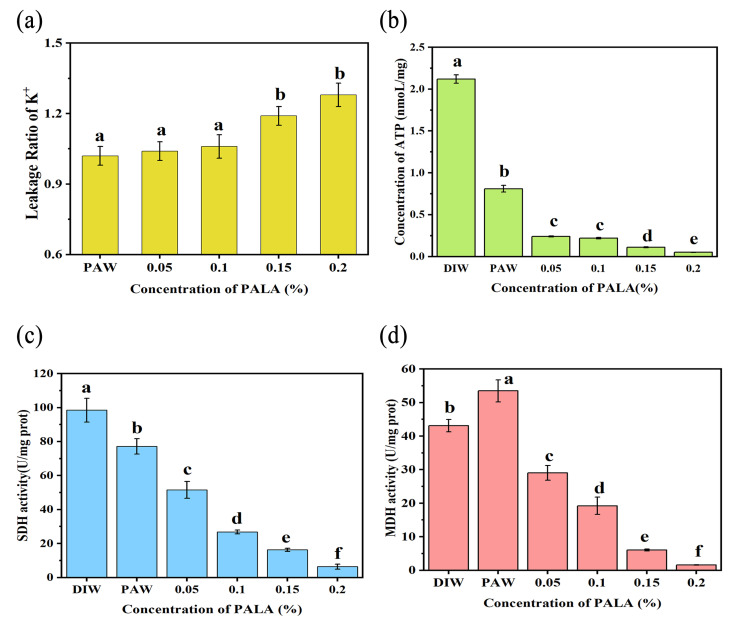
Effect of treatment with DIW, PAW, PALA (0.05%), PALA (0.1%), PALA (0.15%) and PALA (0.20%) on the intracellular ROS levels (**a**) and GSH concentration (**b**) of *Pseudomonas Lundensis*. Bars represent standard deviations (*n* = 3). a–f: values with different lowercase letters significantly differ (*p* < 0.05). The effect of the treatment with DIW, PAW, PALA (0.05%), PALA (0.1%), PALA (0.15%), and PALA (0.20%) on the leakage ratio of K^+^ (**a**), intracellular ATP concentrations (**b**), MDH activity (**c**) and SDH activity (**d**) of *P. Lundensis*. Bars represent standard deviations (*n* = 3). a–f: values with different lowercase letters significantly differ (*p* < 0.05).

**Figure 7 foods-12-01605-f007:**
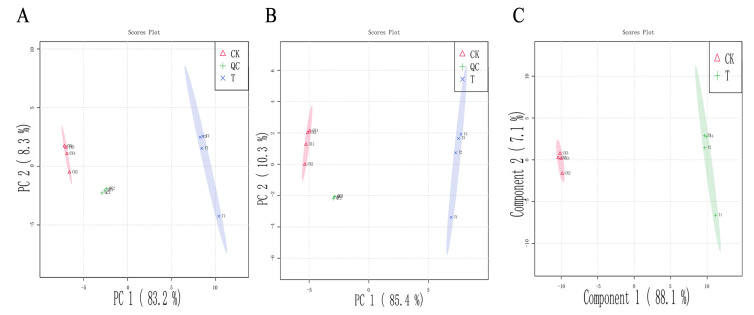
Multivariate cluster analyses of metabolic profiles of treated and control groups. (**A**): PCA plots with positive mode. (**B**) PCA plots with negative mode. (**C**) PLS-DA score plots. CK and T refer to DIW and PALA (0.2%) treatment, respectively.

**Figure 8 foods-12-01605-f008:**
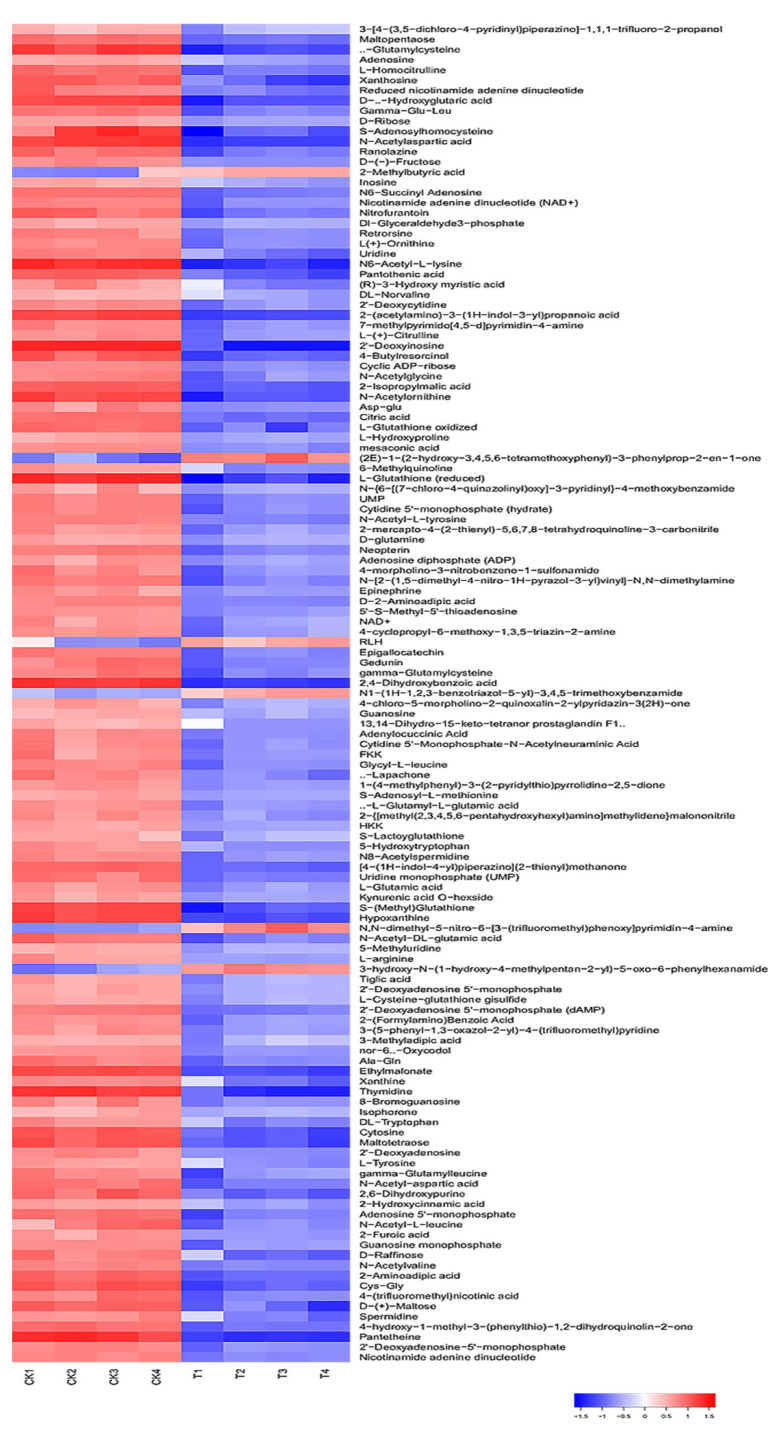
Heat map of differential metabolites cluster. Each row represents one differential metabolite identified. The color represents the metabolite abundance, the red areas in the heat map represent the higher content of metabolites, and the blue areas are the lower concentrations. CK and T refer to DIW and PALA (0.2%) treatment, respectively.

**Figure 9 foods-12-01605-f009:**
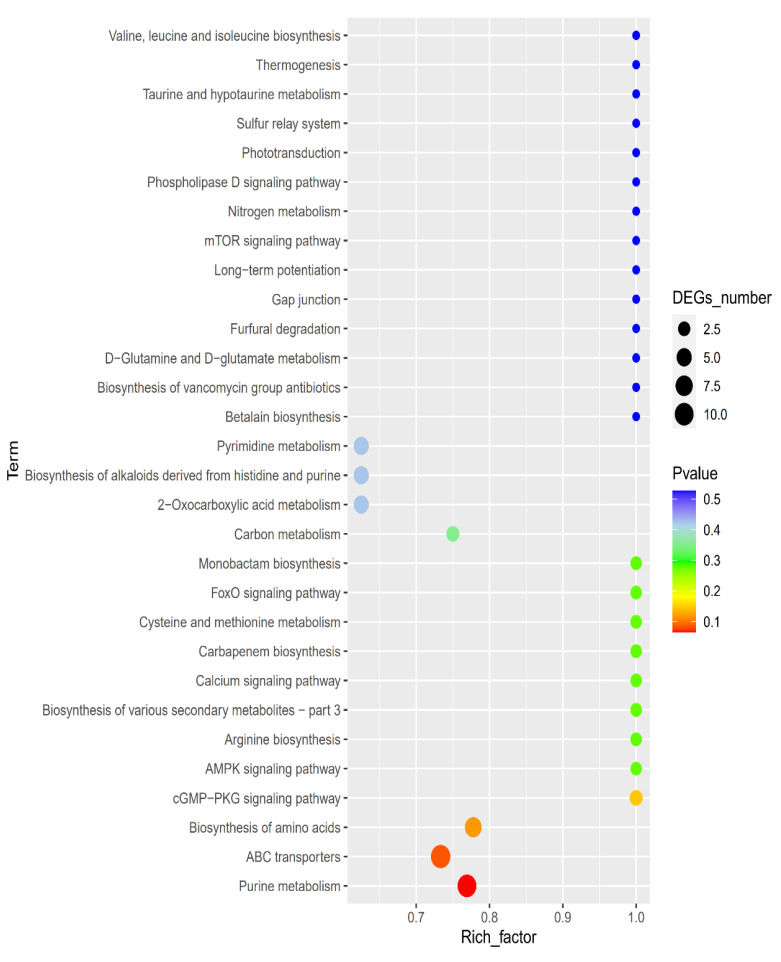
KEGG enrichment map of differential metabolites. Larger bubble plots indicate the higher differential metabolic compound pathway enrichment level at the control and PALA exposures.

**Figure 10 foods-12-01605-f010:**
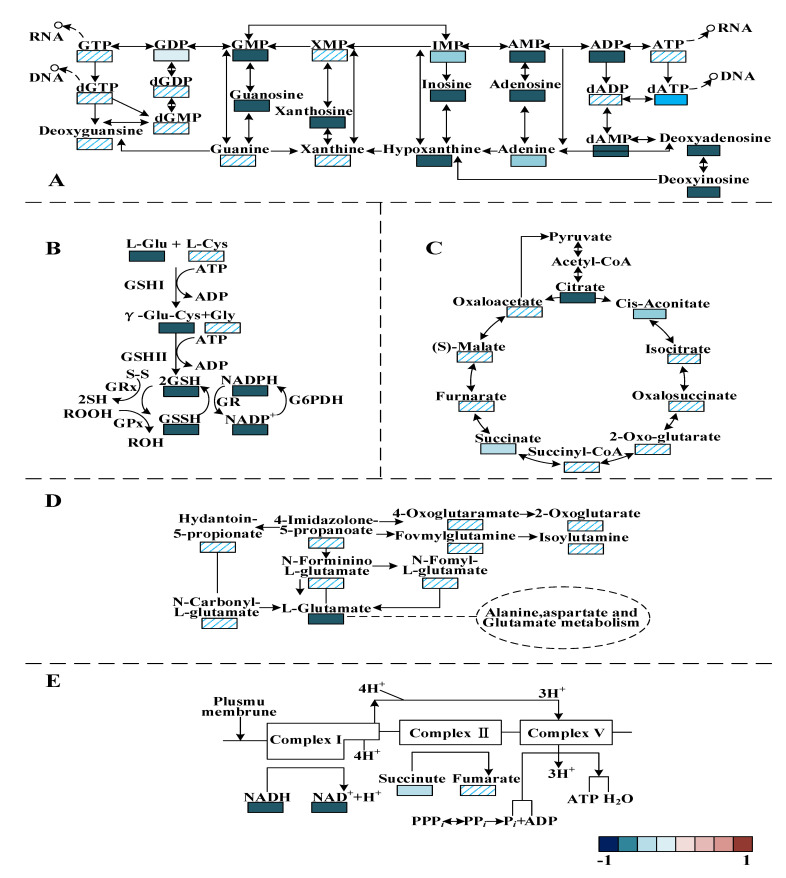
Changes in identified metabolites in response to the PALA exposure in the pathways of purine metabolism (**A**), glutathione (GSH) metabolism (**B**), TCA cycle (**C**), alanine, aspartate and glutamate formation (**D**) and oxidative phosphorylation metabolism (**E**). The colors represent the metabolite abundance ratio of the PALA treatment and the control. Red is upregulated; blue is downregulated.

## Data Availability

The datasets generated for this study are available on request from the corresponding author.

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
