# Peer review of "Unraveling the Antibacterial Mechanism of Plasma-Activated Lactic Acid against Pseudomonas ludensis by Untargeted Metabolomics"

_foods, 2023, doi:10.3390/foods12081605_

Round 1
Reviewer 1 Report
The article entitled “unraveling the antibacterial mechanism of plasma-activated lactic acid against P. ludensi by untargeted metabolomics shows a study on disinfection of meat spoilage bacteria using the plasma-activated lactid acid. The study is still limited in food preservation and reported in clear and good way. Few comments need to addressed as follows
Figure 1. Lg CFU/mL should be changed to “Log CFU/mL”. It is also confusing for the time used for plasma treatment. In The caption, treatment time 0-100 s, but the figure 0-120 s.
In Figure 10, it is difficult to identify the colour. It is suggested to make the figure more clear.
Reviewer 2 Report
Dear Authors,
The manuscript foods-2285911, entiteled 'Unraveling the antibacterial mechanism of Plasma-activated lactic acid against Pseudomonas ludensis by untargeted metabolomics' presents results on the effectiveness of plasma treated lactic acid against some microorganism. The overall manuscript is well written and structured, with good examples and well discussed results. The figures are of good quality but should be improved.
Even if it is well written, the figures need improvement as size and addnotations size:
- Fig1, on page 6, the addnotations inside figure must be made bigger, as manuscript text size, same for figures 2 on page 7. Their figure size should be mage 2x greater.
- Fig 3 page 7 should be 2x grater, and to have a scale bare inside;
- Fig 4, on page 8, bigger, 2x;
- same for Fig5 , Fig6 and 7, also bigger addnotations;
- Fig 8 on page 12 should be mage bigger, whole page hight, also fig 9;
- Fig 10 on page 15 would look better if made wide as page width.
There are nice works presented from the literature, with good examples that are well presented.
Reviewer 3 Report
The provided manuscript aims to determine the antibacterial mechanism of plasma-activated lactic acid against Pseudomonas ludensis. The authors used a variety of approaches - from the visual determination of the antibacterial effect (scanning and transmission electron microscopy) to determining the principle of action such as glutathione levels, reactive oxygen species, malate dehydrogenase, succinic dehydrogenase, and intracellular ATP level and metabolomics analysis. The study is interesting, but I have some negative comments and recommendations:
1) Spaces must be entered in many places in the text!
2) Lines 91-100: Which controls did you use for proving of Pseudomonas lundensis (such as primers, control strain, PCR program, etc.)? What are these biochemical tests? How are you going to convince us that this bacterium belongs to the pseudomonad species? Does it have any number, does it belong in anyone's collection?
3) Lines 135 and 262, 263: The names of bacteria – in Itallic!
4) Lines 290-298, 339-341, 375-376, etc.: These are not results and the place of this text is not here! This is Discussion! Either separate the discussion from the Results or rename the Section!
5) Nothing is visible in Figure 3 and Figure 8!
6) Line 539: The link is missing!
Once you provide me with the necessary information on how you have proven the isolated bacterial species to belong exactly to the Pseudomonas lundensis and provide me Figures 2 and 8 then I will be able to give my final assessment!
Round 2
Reviewer 3 Report
Dear Zhaobin Wang,
Thank you for the answers and clarifications! I recommend the publication of the manuscript!